# Study on the Initiation of Interface Crack in Rock Joints

**DOI:** 10.3390/ma15144881

**Published:** 2022-07-13

**Authors:** Xin Chen, Wei Gao, Shuangshuang Ge, Cong Zhou

**Affiliations:** Key Laboratory of Ministry of Education for Geomechanics and Embankment Engineering, College of Civil and Transportation Engineering, Hohai University, Nanjing 210098, China; 180204010003@hhu.edu.cn (X.C.); 200204010005@hhu.edu.cn (S.G.); czhou1996@163.com (C.Z.)

**Keywords:** interface crack, fracture criterion, rock joints, interfacial fracture, T-stress

## Abstract

The interfacial fracture of rock joints is an important although easily ignored issue in jointed rock engineering. To conduct this study, an interface crack model of rock joints was proposed. By analyzing the ratio of stress intensity factor to fracture toughness, the fracture mode of the interface crack was studied. Based on the Mohr-Coulomb criterion, an interface fracture criterion considering T-stress was established. To verify the proposed fracture criterion, laboratory and numerical tests were conducted. Finally, the effect of relative critical size α, internal friction angle *φ* and cohesion *c* on the initiation of an interface crack was comprehensively discussed. It is concluded that the proposed fracture criterion can predit the initiation of the interface cracks properly. With an increase in cohesion *c*, mode II fracture toughness *K*_II*C*_ also clearly increases. When the absolute value of *K*_I_ is small, the effect of *α* is much larger than that of *φ*. In addition, with an increase in the absolute value of the mode I stress intensity factor, the *φ* of the joint plays a more important role in the initiation of the interface crack.

## 1. Introduction

Natural rock masses are composed of a rock matrix and discontinuities. The existence of discontinuities, such as rock joints, has a significant influence on the strength and deformation properties of rock masses. Generally, it can be found that the prime failure modes of rock masses are the shear slips along the rock joints [1,2]. There are also often flaws inside the rock joints, which have a serious influence on the shear mechanical properties of rock joints [3]. To better understand the failure process of rock joints caused by cracks, the mesomechanics method (i.e., fracture mechanics) should be adopted, as it has been used in rock mechanics [4,5,6]. Therefore, the study on the interfacial fracture of rock joints has an important theoretical and practical significance.

Until now, there have been three methods to study the failure of rock joints, namely laboratory experiments, numerical modeling, and theoretical research. Experimental research tends to focus on the influence of the filling material, surface roughness or matedness of the contact surfaces and the coupling of external environments on the failure of rock joints by laboratory tests. For example, She and Sun [7] conducted compressive shear tests on cement-filled joints and investigated the peak shear strength of these joints. Tang and Wong [8] conducted direct shear tests on artificial rock joints with different contact states and analyzed the variation of the peak shear strength. Zhao et al. [9] carried out several groups of direct shear tests on wetting-treated rock joint specimens and studied their shear failure behavior. However, large numbers of tests are time-consuming and it is a difficult to create some specific conditions in laboratory tests, such as setting the interface crack in the rock joint. Moreover, this kind of study can only reveal some common macro-mechanical properties of the rock joints.

Meanwhile, different from experimental technique, numerical simulation is a suitable method to investigate the mesoscopic failure of rock joints. For example, Guo and Qi [10] simulated rock specimens with unfilled joints using the finite difference method and investigated the progressive failure mode of jointed rocks with various undulate joints from a mesoscopic view. Tian et al. [11] proposed an interface model to simulate the shear failure of cemented concrete–rock joints using the finite element method and analyzed the effect of gradual bond failure processes of cohesive interface elements on the post-peak performance of joints. Park and Song [12] simulated the direct shear test to study the influence of meso-parameters on the shear failure of rock joints based on the bonded particle model. Asadi et al. [13] carried out the shear test simulations on rough rock joints using a particle flow code and studied the effects of the meso-properties as well as the geometrical features on the fracture shear behavior. Compared with the continuum models, the discontinuous models, such as the discrete element method, have some unique characteristics, especially the advantage of simulating crack propagation [14]. Although the failure of rock joints have been investigated from a mesoscopic perspective, the effect of an interface crack in the rock joints, which is the main factor in joint failure, has not been researched.

Theoretical research is a key aspect to the study of the failure of rock joints. So far, the effect of surface roughness of joint surfaces on the failure of rock joints [15,16,17] or establishing strength model which consider various physical parameters (e.g., dilation angle and morphology of rock joints) [18,19,20] is hot topic of the theoretical research. However, generally, in these studies, a semiempirical method has been used, which cannot explain the failure of the joint interface from the mesomechanics. In fact, the failure of rock joints always starts at its positions of weakness (i.e., flaws), which can be studied to good effect by fracture mechanics. To study the initiation mechanism of cracks, there are three main theories: maximum tangential stress criterion [21], strain energy density criterion [22], and energy release rate criterion [23]. Among these, the maximum tangential stress criterion (MTS) is the most widely used in rock material for its simple form and precise physical meaning [24,25,26]. In order to predict crack initiation angle more accurately, T-stress near crack tip was introduced to the MTS criterion [27]. The initiation of a crack which is in single medium can be well solved by the theories above-mentioned. When a crack is in the rock joint interface, which is termed as interface crack, the theories are difficult to apply to such case due to the discontinuity of strength parameters along the joint. For interface cracks, the related theoretical research is mainly focused on the interface crack of composite materials (bimaterials) with strong interface, for which the stress field at the crack tip exhibits oscillatory singularity induced by mismatching of the dissimilar materials on both sides of the interface [28,29]. For example, in Cartesian coordinates, Deng [30] first derived analytical solution of the crack-tip stress field for the dissimilar materials. Zhou and Li [31] obtained full elastic fields of the interface crack tip and corresponding energy release rate for orthotropic bi-material. Banks-Sills [32] presented failure criteria for a crack between two dissimilar materials based on energy release rate. Mega et al. [33] proposed several two- and three-dimensional mixed-mode interface failure criteria for predicting delamination failure in multidirectional, laminate composites. Zhang et al. [34] proposed a modified critical energy density criterion for peridynamic interface bond failure analysis. In fact, for the cracks in the rock joint interface, both sides of the interface are the same materials and there is a fracture process zone at the crack tip of rock-like materials [35], for which classical interfacial fracture mechanics is also unsuitable for determining the fracture of the interface crack of rock joints. Moreover, the strength parameters of the rock joint interface are much lower than that of rock matrix, which is termed as weak interface. The weak interface will affect the occurrence of mode I fracture [36] which is always occurred as the cracks is in rock matrix [37]. Therefore, establishing a fracture criterion for the interface crack in the rock joint is very urgent. 

In this paper, the interface crack model of rock joints is proposed for the first time. Moreover, by using fracture mechanics, the fracture mode of the interface crack is determined, and a new fracture criterion for interface cracks of rock joints is proposed. Lastly, the new fracture criterion is verified by the experimental and numerical tests using the discrete element method.

## 2. Interface Crack Model

Generally, there are many defects (weaknesses) in the rock joints. Hence, a model of a rock joint containing a pre-existing crack is shown as in Figure 1. In this model, for joints with a small thickness, the joint can be simplified to a mathematical interface where the interface crack exists.

For the interface crack, Deng [30] derived general expressions of the crack-tip stress field for dissimilar materials, which is given as:(1)σ(r,θ)=∑n=0∞rn−12{Re(ηnIknriεn)2πσ^nI(θ)+Im(ηnIIknriεn)2πσ^nII(θ)+k3n2πσ^nIII(θ)}
where *ε* is the oscillation index, *r* and *θ* are polar coordinates when the ordinate origin is assumed to be at the crack tip, and *k_n_*, $\eta_{n}^{I}$, $\eta_{n}^{II}$, ${\hat{\sigma }}_{n}^{I}\left(\theta \right)$, ${\hat{\sigma }}_{n}^{II}\left(\theta \right)$ and ${\hat{\sigma }}_{n}^{III}\left(\theta \right)$ are expressions in the derivation. Due to the complexity of the expressions for these parameters, the specific expressions are not listed here. The details can be found in [30].

For bimaterials, the interface crack-tip stress field exhibits oscillatory features. However, for rock joint, both sides of the joint interface are the same materials. Therefore, the bimaterial becomes homogeneous, which is illustrated by *ε* = 0. Crack-tip stress fields with oscillatory features are simplified to those of traditional fracture mechanics problems. Therefore, the stress field at the interface crack tip of the joint is as follows [38]:(2)$$\{\begin{array}{l} \sigma_{x}\left(r,\theta \right)=\frac{K_{I}}{\sqrt{2\pi r}}\cos\frac{\theta }{2}\left(1-\sin\frac{\theta }{2}\sin\frac{3\theta }{2}\right)-\frac{K_{II}}{\sqrt{2\pi r}}\sin\frac{\theta }{2}\left(2+\cos\frac{\theta }{2}\cos\frac{3\theta }{2}\right)\\ \sigma_{y}\left(r,\theta \right)=\frac{K_{I}}{\sqrt{2\pi r}}\cos\frac{\theta }{2}\left(1+\sin\frac{\theta }{2}\sin\frac{3\theta }{2}\right)+\frac{K_{II}}{\sqrt{2\pi r}}\sin\frac{\theta }{2}\cos\frac{\theta }{2}\cos\frac{3\theta }{2}\\ \tau_{xy}\left(r,\theta \right)=\frac{K_{I}}{\sqrt{2\pi r}}\cos\frac{\theta }{2}\sin\frac{\theta }{2}\cos\frac{3\theta }{2}+\frac{K_{II}}{\sqrt{2\pi r}}\cos\frac{\theta }{2}\left(1-\sin\frac{\theta }{2}\cos\frac{3\theta }{2}\right) \end{array}$$
where *K*_I_ and *K*_II_ are the stress intensity factors for crack mode I and II, respectively.

The interface crack model is shown in Figure 1. In Figure 1, the stress on a crack surface whose dip angle is *β* can be described as:(3)$$\{\begin{array}{l} \sigma_{y}=\frac{\sigma \left(1+k\right)}{2}+\frac{\sigma \left(1-k\right)}{2}\cos2\beta \\ \tau_{xy}=\frac{\sigma \left(1-k\right)}{2}\sin2\beta \end{array}$$
where *k* is the ratio of minimum principal stress to maximum principal stress, and *σ_y_* and *τ_xy_* are the normal and shear stresses of the joint plane, respectively.

Then, according to the basic theory of fracture mechanics [38], the stress intensity factors of mode I and II cracks are given as
(4)$$\{\begin{array}{l} K_{I}=-\sigma_{y}\sqrt{\pi a}\\ K_{II}=-\tau_{xy}\sqrt{\pi a} \end{array}$$
where a is the half-length of the crack.

In Equations (3) and (4), *σ_y_* is a positive value, which represents the tensile stress. Under compression conditions, the value of the stress intensity factor *K*_I_ should be negative, which illustrates that *K*_I_ is able to restrain the initiation of cracks [39].

## 3. Initiation of Interface Crack

### 3.1. Determination of Crack Initiation Angle

According to the maximum circumferential stress criterion, the crack will initiate along the direction of maximum tensile stress. However, for interface crack in rock joints, the crack exists in a joint plane with a strength much lower than that of rock block [40], which means that the fracture toughness of the joint plane is much weaker than that of rock block. Therefore, the rock near the interface crack tip cannot be regarded as homogeneous material. In this study, the fracture toughness of mode I and mode II fractures in the direction of joint extension (*θ* = 0) is defined as $K_{IC\_joint}^{}$ and $K_{IIC\_joint}^{}$, respectively, and as $K_{IC\_matrix}^{}$ and $K_{IIC\_matrix}^{}$ for the other directions (*θ* ≠ 0), respectively.

When the interface crack is under biaxial compression, the stress field at the crack tip in the polar coordinate can be described as [38]:(5)$$\{\begin{array}{l} \sigma_{r}=\frac{1}{2\sqrt{2\pi r}}\left[K_{I}\left(3-\cos\theta \right)\cdot \cos\frac{\theta }{2}+K_{II}\left(3\cos\theta -1\right)\cdot \sin\frac{\theta }{2}\right]\\ \sigma_{\theta }=\frac{1}{2\sqrt{2\pi r}}\cos\frac{\theta }{2}\left[K_{I}\left(1+\cos\theta \right)-3K_{II}\sin\theta \right]\\ \tau_{r\theta }=\frac{1}{2\sqrt{2\pi r}}\cos\frac{\theta }{2}\left[K_{I}\sin\theta +K_{II}\left(3\cos\theta -1\right)\right] \end{array}$$

The fracture modes (here only mode I and mode II fractures are considered) depend on *σ_θ_* and *τ_r_**_θ_*, regardless of which kind of stress is applied [41]. Accordingly, whether mode I or II fractures occur at the initiation angle *θ*_I*C*_ or *θ*_II*C*_ depends on the critical value of the circumferential tensile stress *σ_θ_* or the shear stress *τ_r_**_θ_*. Here, different joint dip angles *β* (15°, 30°, 45°, 60°) and ratios of minimum principal stress to maximum principal stress *k* (0, 0.1, 0.3, 0.5, 0.7) were chosen in order to analyze the initiation condition of the interface crack.

In order to better analyze the relationship between stress intensity factors and fracture toughness, the stress intensity factors in different direction angles *θ* near the crack tip can be defined as [41]:(6)$$\{\begin{array}{c} K_{I}\left(\theta \right)=\underset{r\to 0}{\lim}\sigma_{\theta }\sqrt{2\pi r}\\ K_{II}\left(\theta \right)=\underset{r\to 0}{\lim}\tau_{r\theta }\sqrt{2\pi r} \end{array}$$

Equation (5) was substituted into Equation (6), resulting in:(7)$$\{\begin{array}{c} K_{I}\left(\theta \right)=\frac{1}{2}\cos\frac{\theta }{2}\left[K_{I}\left(1+\cos\theta \right)-3K_{II}\sin\theta \right]\\ K_{II}\left(\theta \right)=\frac{1}{2}\cos\frac{\theta }{2}\left[K_{I}\sin\theta +K_{II}\left(3\cos\theta -1\right)\right] \end{array}$$

Whether a mode I or mode II fracture occurs depends on whether *K*_I max_ or *K*_II max_ takes precedence to reach its critical value, *K*_I*C*_ or *K*_II*C*_ [41]. Therefore, the fracture mode can be determined by:(8)$$\frac{K_{I\left(II\right)\max}}{K_{I\left(II\right)C}}=1$$
where *K*_I(II)__max_ is the maximum of the mode I or II stress intensity factor around the crack tip.

The $K_{IIC}/K_{IC}$ ratio is about 2.6 [41] and the fracture toughness in other directions (*θ* ≠ 0) is about 4 times that in the direction of joint extension (*θ* = 0) [42]. In order to determine the initiation angle and the representative conclusions, the ratio of fracture toughness in other directions and that in the direction of joint extension ($\frac{K_{I\left(II\right)C\_matrix}^{}}{K_{I\left(II\right)C\_joint}^{}}$) was selected to be 4. The variation in $\frac{K_{I\left(II\right)}\left(\theta \right)}{K_{I\left(II\right)C}}$ in various conditions was calculated as in Figure 2. It is noted that to better facilitate the calculation and illustrate the results, the non-dimension of $K_{I\left(II\right)}\left(\theta \right)$ (i.e., $K_{I\left(II\right)}\left(\theta \right)/\sigma \sqrt{\pi a}$) was used to analyze the stress intensity factor distribution and mode I fracture toughness of rock matrix $K_{IC\_matrix}^{}$ was set set as 1.

As can be seen from Figure 2a–d, when *k* is set to 0.1 for the mode I stress intensity factor, the maximum value increases first and then decreases with an increase in the crack dip angle *β* (the angle between the crack and horizontal direction). The angle (*θ*) corresponding to the maximum value gradually decreases with increase in the crack dip angle *β*, which is in agreement with the experiments by Rao et al. [41]. Although the mode I stress intensity factor reaches its maximum when the crack dip angle *β* is about 60°, the mode II fracture still occurs due to a lower fracture toughness in the joint plane. From Figure 2c,e–h, it is concluded that the mode I stress intensity factor maximum decreases as confining pressure increase and the mode I stress intensity factor is negative in all directions when k is equal to 0.7, which illustrates that the existence of confining pressure inhibits the growth of the mode I stress intensity factor. Considering the above analysis, the conditions such that the crack dip angle *β* is about 60° and the confining pressure is 0 are most suitable for a mode I fracture to occur. However, as shown in Figure 2e, the maximum of $K_{I}\left(\theta \right)/K_{IC}$ and $\left|K_{II}\left(\theta \right)\right|/K_{IIC}$ are 0.3779 and 0.6662, respectively. According to Equation (8), a mode II fracture will still occur and the interface crack will initiate along the joint plane.

Therefore, for interface cracks in rock joints, a mode II fracture often occurs, which is different from traditional rock fracture mechanics, where a mode I fracture more commonly occurs and the initiation angle is about 70.5° [43].

### 3.2. Crack Initiation Criterion

Since a mode II fracture generally occur for interface cracks in rock joints, the crack initiation stress cannot be determined by the maximum circumferential stress criterion. Therefore, in this study, a new fracture criterion is proposed, which is used to predict the initiation stress of the interface crack under compression-shear conditions.

When traditional fracture mechanics theory is used to study the crack initiation mechanism of rock, the Williams expansion is frequently used, which can be written as [44]:(9)$$\sigma_{ij}=A_{1}r^{-1/2}f_{ij}^{1}\left(\theta \right)+A_{2}f_{ij}^{2}\left(\theta \right)+A_{3}r^{1/2}f_{ij}^{3}\left(\theta \right)+\cdots$$
where *r* and *θ* are polar coordinates when the ordinate origin is assumed to be at the crack tip. In the first term, *A*_1_ can be regarded as the stress intensity factors of *K*_I_ or *K*_II_ when the constant terms are absorbed into the trigonometric function $f_{ij}^{n}$ terms. In the second term, *A*_2_ represents the stress that is directly applied to the crack line on the normal plane. The third term is the high order term.

In traditional fracture mechanics theory, only the singular stress term *r*^−1/2^ in Williams expansions is considered, which incurs a difference between the theoretical and experimental results [45]. Moreover, for rock material, there is a fracture process zone at the crack tip [46]. For the stress field of the fracture process zone at the crack tip, the proportion of singular stress term decreases and the T-stress term increases [47]. Thus, the effect of T-stress should be considered.

Based on Equation (5), when T-stress is considered, the following is derived:(10)$$\{\begin{array}{l} \sigma_{r}=\frac{1}{2\sqrt{2\pi r}}\left[K_{I}\left(3-\cos\theta \right)+K_{II}\left(3\cos\theta -1\right)\right]+T_{x}\cos^{2}\theta +T_{y}\sin^{2}\theta \\ \sigma_{\theta }=\frac{1}{2\sqrt{2\pi r}}\cos\frac{\theta }{2}\left[K_{I}\left(1+\cos\theta \right)-3K_{II}\sin\theta \right]+T_{x}\sin^{2}\theta +T_{y}\cos^{2}\theta \\ \tau_{r\theta }=\frac{1}{2\sqrt{2\pi r}}\cos\frac{\theta }{2}\left[K_{I}\sin\theta +K_{II}\left(3\cos\theta -1\right)\right]+\frac{1}{2}\left(T_{y}-T_{x}\right)\sin\left(2\theta \right) \end{array}$$

The T-stress can be described as [45]:(11)$$\{\begin{array}{c} T_{x}=\sigma \left(\cos^{2}\beta +k\sin^{2}\beta \right)\\ T_{y}=\sigma \left(\sin^{2}\beta +k\cos^{2}\beta \right) \end{array}$$

According to the above analysis, for the interface crack, a “shear fracture” (i.e., mode II fracture) along the joint will generally occur. Moreover, the initiation stress of the interface crack is determined by the stress states of the fracture process zone [48], which illustrates that the initiation stress is a function of the stress in the fracture process zone. Therefore, the Mohr–Coulomb criterion, which is used to describe the shear failure of materials, can be adopted to describe the initiation of the interface crack. The formula can be written as
(12)τ=c+σtanφ
where *c* is the cohesion, *φ* is the internal friction angle, *τ* is the shear stress and *σ* is the normal stress of joints.

Equation (10) is substituted into Equation (12) and *θ* set as equal to 0 (the initiation angle of the interface crack in the rock joint is 0°), deriving the following:(13)$$\frac{K_{II}}{\sqrt{2\pi r}}+\tan\varphi \left[\frac{K_{I}}{\sqrt{2\pi r}}+T_{y}\right]=c$$

Both sides of Equation (13) are multiplied by $\sqrt{2\pi r}$, and the following equation can be obtained:(14)$$K_{II}+\tan\varphi \left(K_{I}+T_{y}\sqrt{2\pi r}\right)=c\sqrt{2\pi r}$$

For Equation (14), *r* = *r_c_* and, for simplification, the critical radius *r_c_* is presented in the dimensionless form of $\alpha =\sqrt{2r_{c}/a}$. Moreover, the stress field of the fracture process zone is considered. Then, Equation (11) is substituted into Equation (14), deriving:(15)$$K_{II}+\tan\varphi \left(1+\alpha \right)K_{I}=\alpha c\sqrt{\pi a}$$

In Equation (15), the term $c\sqrt{\pi a}$ is consistent with the definition of stress intensity factor in form, and as such is the mode II fracture toughness produced by the cohesion *c*. When the average mode II fracture toughness of the fracture process zone is considered, the following is obtained:(16)$$K_{IIC}=c\sqrt{2\pi r_{c}}$$

Substituting $\alpha =\sqrt{2r_{c}/a}$ into the right side of Equation (15) and combining with Equation (16) obtains:(17)$$\alpha c\sqrt{\pi a}=\sqrt{2r_{c}/a}\cdot c\sqrt{\pi a}=c\sqrt{2\pi r_{c}}=K_{IIC}$$

Equation (17) is substituted into Equation (15), and the fracture criterion of the interface crack can be obtained:(18)$$K_{II}+\tan\varphi \left(1+\alpha \right)K_{I}=K_{IIC}$$

## 4. Verification Study

### 4.1. Verification by Experimental Study

Tan and Xu [49] conducted double-edge notched single-edge compression tests (DNSCTs) on the layered concrete, as shown in Figure 3. In their study, the influence of the pouring interval of two layers on the bond characteristics was investigated. For specimens with the same pouring interval, the cohesion *c* can be determined by direct shear tests and the mode II fracture toughness can be determined by the DNSCT. For the specimens with different pouring intervals, the relationship of *K*_II*C*_ and *c* is shown in Figure 4. As shown in Figure 4, there is a linear relationship between K_II*C*_ and *c*, where the fitting coefficient *R*^2^ is 0.9980.

In this test, the interface crack is in a pure shearing state. For this condition, the proposed fracture criterion can be generated as the following expression:(19)$$K_{II}=K_{IIC}=c\sqrt{2\pi r_{c}}$$

From the above equation (Equation (19)), the theoretical relationship between *K*_II*C*_ and *c* can be obtained. Therefore, the theoretical result is in good agreement with the real test result. Hence, the criterion can be verified by real tests.

Generally, it is difficult to control the test conditions of mode II fractures. Therefore, it is very challenging to obtain the mode II fracture toughness from real tests. However, by using the theoretical study in this paper, the quantitative relation between mode II fracture toughness and the shear strength parameter *c* is determined, avoiding the difficulty of conducting the real test.

### 4.2. Verification by the Numerical Study

Due to the difficulty of conducting the test in the laboratory using the interface crack specimens, to comprehensively and thoroughly verify the proposed fracture criterion of the interface crack, a numerical test is used here.

The numerical simulation material is siltstone [50]. The numerical model is generated by particle flow code (PFC2D), and the contact model between particles is the bonded-particle model (BPM). The size of the two-dimensional model is such that the width is 50 mm and the height is 100 mm. There is a consecutive joint in the model. For a comprehensive study, four models, for which the inclination angles of the joint are 35.8°, 45°, 54.2°, and 63.4° from the horizontal, have been constructed, as shown in Figure 5. The interface crack is in the middle of the joint, which has a length of 12 mm. The particle diameters of these models satisfy a uniform distribution, with a range between 0.17 mm and 0.3 mm. For the different numerical models, there are different numbers of particles, contacts, etc. For example, for the numerical model with a joint angle of 45°, there are 28,296 particles, 76,451 contacts, 75,946 parallel bonds and 289 smooth-joint contacts. The particle density of siltstone is 2400 kg/m^3^, and the average porosity is 7%, which is the same as the physical parameters of real specimens [50].

In this study, the smooth-joint contact model (SJM) is used to simulate the behavior of the joints. As shown in Figure 6, the SJM suggests that the particles will overlap with each other along the smooth interface. Hence, this contact model can simulate the behavior of the joints more realistically, which avoids the unrealistic mechanicals properties of joints induced by the bumpiness or roughness of the interface surfaces [51]. Therefore, in this way, the initiation of an interface crack under uniaxial compression can be simulated accurately.

#### 4.2.1. Calibration of Micro-Parameters

For the particle flow code, micro-parameters are used and their calibration is crucial. In order to simulate the behavior of single-jointed rock, the numerical model of the single-jointed rock specimen is calibrated based on the stress-stain curve obtained by a real experiment [50], as shown in Figure 7. From Figure 7, it is clear that the mechanical behavior of numerical model is in agreement with that of the real specimen. For the single-jointedrock specimen, the joint inclination angle is 68°. The calibrated micro-parameters of the numerical model are listed in Table 1.

#### 4.2.2. Results of Numerical Study

To reproduce the real static response of the specimen, the displacement loading from the upper wall is applied, which simulates the lab uniaxial compression. By fixing the bottom wall and moving the upper wall down, causing an increase in upper loading, the initiation of the interface crack can be clearly simulated. According to [52], the quasi-static loading rate can be set to 0.1 × 10^−3^ mm/min. By using numerical studies, the initiation stress of the interface crack specimens can be obtained, as shown in Figure 8. It should be noted that the initiation stress is determined when the first fractures occur around the crack tip. In Figure 8, the red lines denote the fractures at the interface crack tip. The initiation stress of the interface crack with different joint inclination angles are 55.23 MPa for *β* = 35.8°, 28.28 MPa for *β* = 45°, 19.50 MPa for *β* = 54.2° and 20.01 MPa for *β* = 63.4°. The corresponding mode I and II stress intensity factors can be calculated based on Equations (3) and (4), as listed in Table 2.

Liu [45] pointed out that for metal and polymethyl methacrylate materials, the test results are ideal when *α* is equal to 0.1. For rock material, the critical radius *r*_c_ is larger than that of metal and polymethyl methacrylate material and related research [47] recommended that relative critical size *α* of less than 1 is appropriate. Therefore, the proposed fracture criterion with a varied relative critical size *α*, which range from 0.1 to 0.9, are compared with the numerical results. The mode I and II stress intensity factors and the interface crack fracture criterion with a varied relative critical size *α* are summarized in Figure 9.

As shown from Figure 9, based on the new fracture criterion, the theoretical curves (*μ* = tan20° = 0.36) are linear. As the relative critical size α increases, the slope and intercept increase monotonically, which illustrates that the mode II fracture toughness and initiation stress of the interface crack increase. When *α* = 0.6, the numerical test results are in better agreement with the theoretical results. Moreover, according to Equation (14), the critical size *r_c_* can be determined as 1.08 mm.

## 5. Discussion

In Equation (18), there are three controlling parameters, which are the cohesion *c*, internal friction angle *φ* and relative critical size *α*. The influence of those parameters on the initiation of interface crack is analyzed as follows. Moreover, some comparisons with other studies were also discussed.

### 5.1. Effect of Cohesion c

To study the effect of cohesion *c*, the parameter *c* is taken as 7.7 MPa, 17.7 MPa and 27.7 MPa, and the other parameters, such as internal friction angle *φ* and relative critical size *α*, were fixed as 20° and 0.6, respectively. According to these conditions, the corresponding results are summarized in Figure 10. It can be seen that with the increase in cohesion *c*, the mode II fracture toughness *K*_II*C*_ is 0.87 MPa‧m^0.5^, 1.54 MPa‧m^0.5^ and 2.31 MPa‧m^0.5^, respectively, which illustrates that the mode II fracture toughness *K*_II*C*_ increases with the increase in cohesion. The initiation stress of the interface crack increases with the increase in cohesion. It should be noted that the slope of the relationship curves between mode I and II stress intensity factors remains unchanged, which indicates that there are no differential effects of varying cohesion *c* on the initiation of the interface crack.

In essence, the fracture toughness is the characteristic constant of materials, which represents the resistance to the propagation of a crack [38]. When there is no normal stress on the joint plane (pure shearing), the resistance to the propagation of crack (i.e., mode II fracture toughness) is provided by the cohesion, which is generated by the mutual attraction between the molecules. Therefore, when the cohesion c increases, the mode II fracture toughness increases linearly.

### 5.2. Effect of Internal Friction Angle φ

To study the effect of the friction angle *φ*, parameter *φ* was taken as 20°, 25°, 30° and 35°, and the other parameters, such as cohesion *c* and relative critical size *α*, were fixed as 7.7 MPa and 0.6, respectively. According to these conditions, the corresponding results are summarized in Figure 11. It can be seen from Figure 11 that the relationship curves between mode I and II stress intensity factors monotonically increase, and with the increase in internal friction angle *φ*, the slope of the curves enlarges. However, the intercept term of the curves remains unchanged. This indicates that the internal friction angle affects the interface crack initiation stress and has no influence on the mode II fracture toughness *K*_II*C*_.

The internal friction angle is another shear strength parameter, which represents frictional properties and the shear strength of the contact area [53]. When there is no normal stress on the joint plane, the mode II fracture toughness is provided by the cohesion. Therefore, the mode II fracture toughness *K*_II*C*_ will not be affected by the internal friction angle. When there is normal stress on the joint plane, the mode I stress intensity factor *K*_I_ is negative in the fracture process zone. With an increase in the internal friction angle, more driving force is needed to overcome the frictional strength in the fracture process zone. Therefore, the slope of the relationship curves between mode I and II stress intensity factors increases with the increase in the internal friction angle.

### 5.3. Effect of Relative Critical Size α

To study the effect of relative critical size *α*, parameter α was taken as 0.4, 0.5, 0.6 and 0.7, and the other parameters, such as cohesion *c* and internal friction angle *φ*, were fixed as 7.7 MPa and 20°, respectively. According to these conditions, the corresponding results are summarized in Figure 12. It can be seen from Figure 12 that the relationship curves between mode I and II stress intensity factors monotonically increase, and with the increase in relative critical size *α*, both the slope and intercept of the curves enlarge. Comparing Figure 11 and Figure 12, it can be concluded that when the absolute value of *K*_I_ is small, the effect of *α* on the initiation of the interface crack is much greater than that of *φ*. Then, with the increase in $\left|K_{I}^{}\right|$, the internal friction angle *φ* of the joint plays a more important role in the initiation of the interface crack.

The relative critical size is also regarded as the characteristic constant of materials in the field of rock fracture mechanics [45]. When the relative critical size increases, more driving force is needed to overcome the cohesion and frictional strength. Therefore, both the slope and intercept of the relationship curves between mode I and II stress intensity factors enlarge.

### 5.4. Comparison with Other Studies

To further study the characteristics of this fracture criterion, it was compared with several traditional crack initiation criterion, which are listed in Table 3. The classic MTS criterion [21] and the MTS criterion considering T-stress [45] were chosen for this comparison. 

As shown in Table 3, although the application scope and the fracture mode of the proposed criterion are different from that of the two other traditional crack initiation criterions, the initiation conditions are essentially the same, which depend on whether the absolute value of the stress intensity factor reaches its fracture toughness. Although the classical MTS criterion is concise, the MTS criterion considering T-stress is more accurate in predicting the initiation angle. For the proposed criterion, the number of key parameters is six, which is relatively concise and easy to understand. Moreover, based on the fixed initiation angle, the initiation stress can be predicted well.

## 6. Conclusions

In order to solve the interfacial fracture of rock joints, the interface crack model of joints was proposed for the first time. Moreover, based on the proposed interface crack model, a new fracture criterion considering T-stress was proposed. Then, the proposed fracture criterion was verified by the laboratory tests and numerical study. Finally, the effect of relative critical size *α*, internal friction angle *φ* and cohesion *c* on the initiation of the interface crack was discussed, with the following conclusions drawn:
(1)A Mode II fracture generally occurs in the interface crack of rock joints, which is different from the fracture mode that often occurs in the rock matrix.(2)The theoretical results calculated by the proposed fracture criterion are in good agreement with the experimental and numerical results. Hence, the proposed fracture criterion could be verified by the test results.(3)The effect of T-stress was considered in the fracture criterion. It is shown that the cohesion *c* and internal friction angle *φ* increase with the increase in relative critical size *α*. (4)The relative critical size α, internal friction angle *φ* and cohesion *c* all affect the initiation of the interface crack. With an increase in cohesion *c*, mode II fracture toughness *K*_II*C*_ clearly increases. When the value of *K*_I_ is small, the effect of α on the initiation of the interface crack is much greater than that of *φ*. Then, with an increase in $\left|K_{I}^{}\right|$, the *φ* of the joint plays a more important role on the initiation of the interface crack.

However, the proposed fracture criterion is primary study of the interface fracture in rock joints. Only the case of pure mode II fracture was verified by experiments, and the other conditions were verified by numerical simulations based on experimental data. Therefore, designing more experiments and simulation conditions for the fracture criterion remain challenges for our future research. Moreover, the proposed fracture criterion combined with cohesive zone model can be written into distinct element code modeling, which can be used to analyze the interface fracture of jointed rock in geotechnical engineering.

## Figures and Tables

**Figure 1 materials-15-04881-f001:**
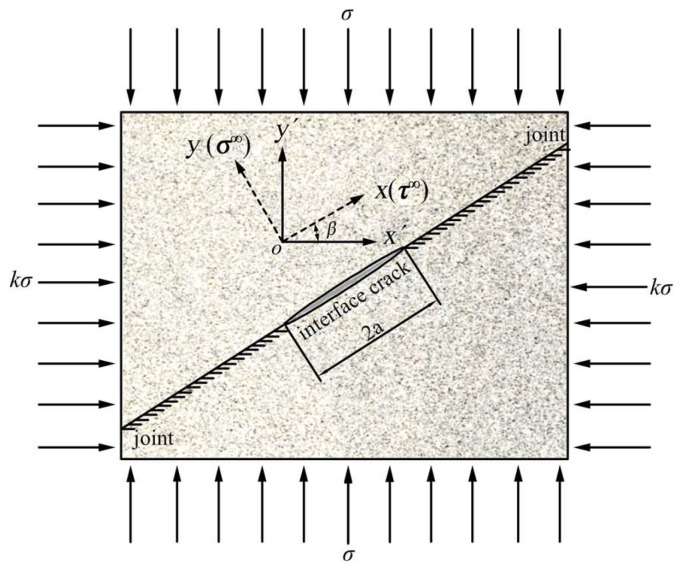
Interface crack model.

**Figure 2 materials-15-04881-f002:**
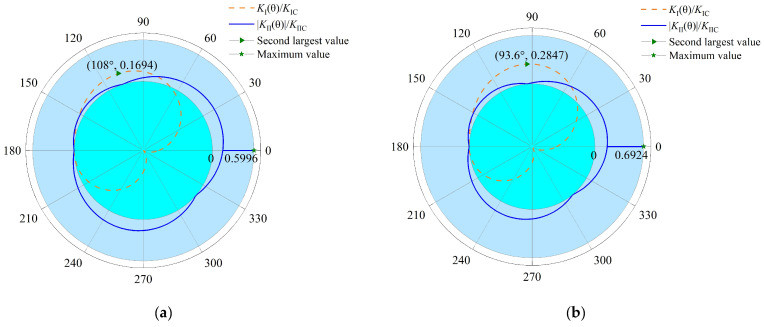
Variation in $K_{I}\left(\theta \right)/K_{IC}$ and $\left|K_{II}\left(\theta \right)\right|/K_{IIC}$ under compression-shear loading. (**a**) *k* = 0.1 *β* = 30°, (**b**) *k* = 0.1 *β* = 45°, (**c**) *k* = 0.1 *β* = 60°, (**d**) *k* = 0.1 *β* = 75°, (**e**) *β* = 60° *k* = 0, (**f**) *β* = 60° *k* = 0.3, (**g**) *β* = 60° *k* = 0.5, (**h**) *β* = 60° *k* = 0.7.

**Figure 3 materials-15-04881-f003:**
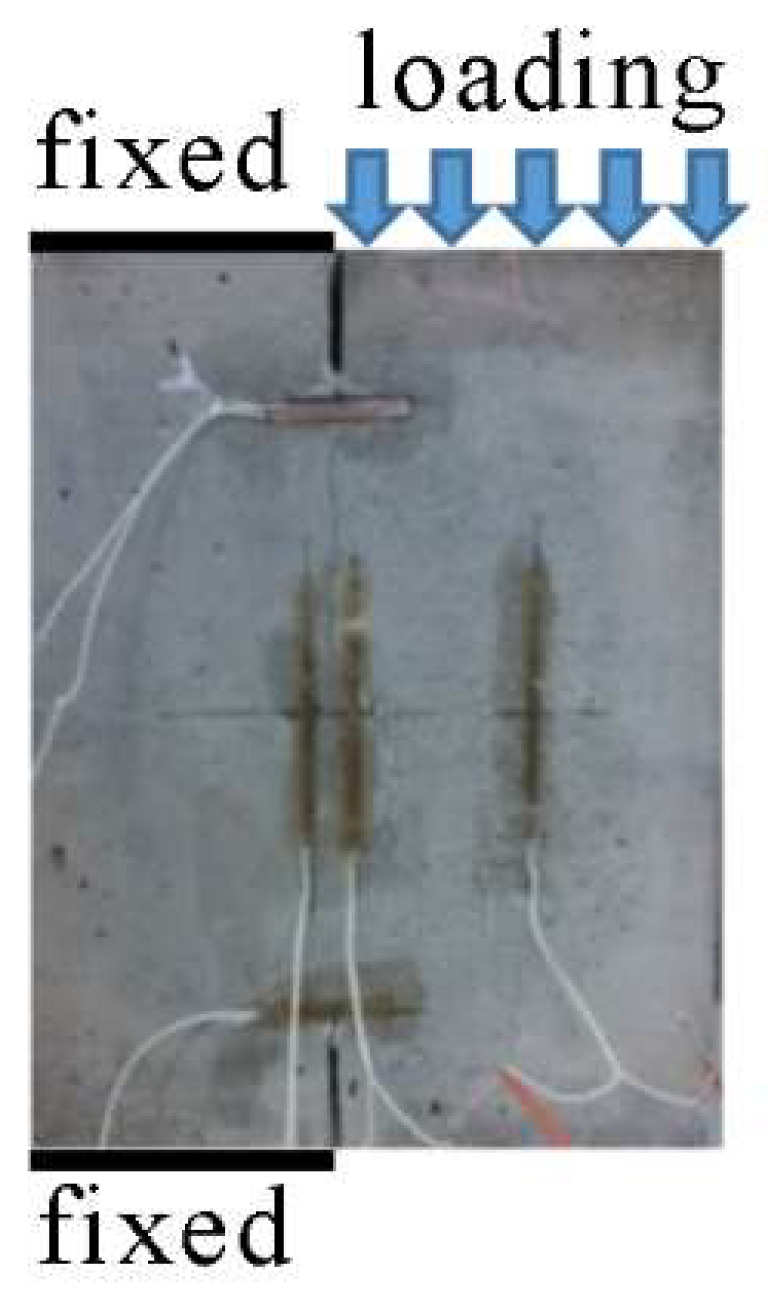
Specimen for DNSCT of mode II shearing [49].

**Figure 4 materials-15-04881-f004:**
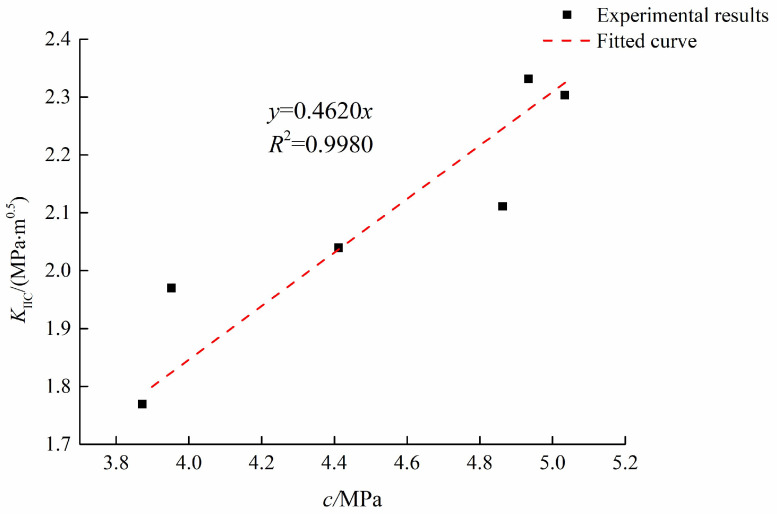
Linear relation of *K*_II*C*_ and *c* (modified from [49]).

**Figure 5 materials-15-04881-f005:**
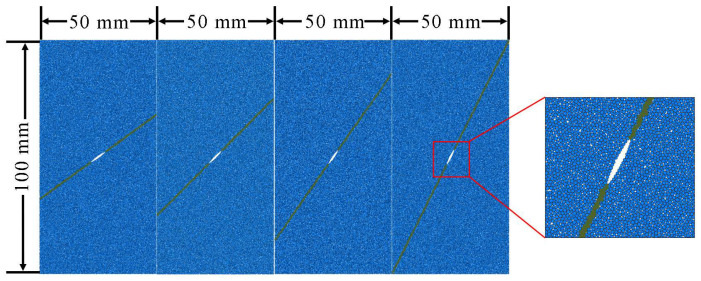
Interface crack specimens with different joint inclination angles (the inclination angles of the joint, from left to right, are: 35.8°, 45°, 54.2°, and 63.4°).

**Figure 6 materials-15-04881-f006:**
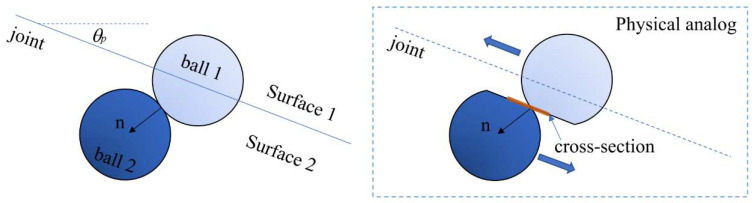
Smooth-joint contact model [51].

**Figure 7 materials-15-04881-f007:**
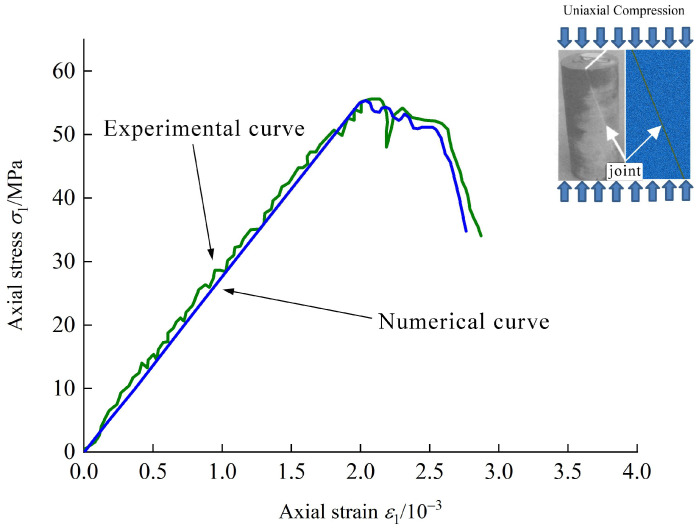
Comparison between numerical and experimental results.

**Figure 8 materials-15-04881-f008:**
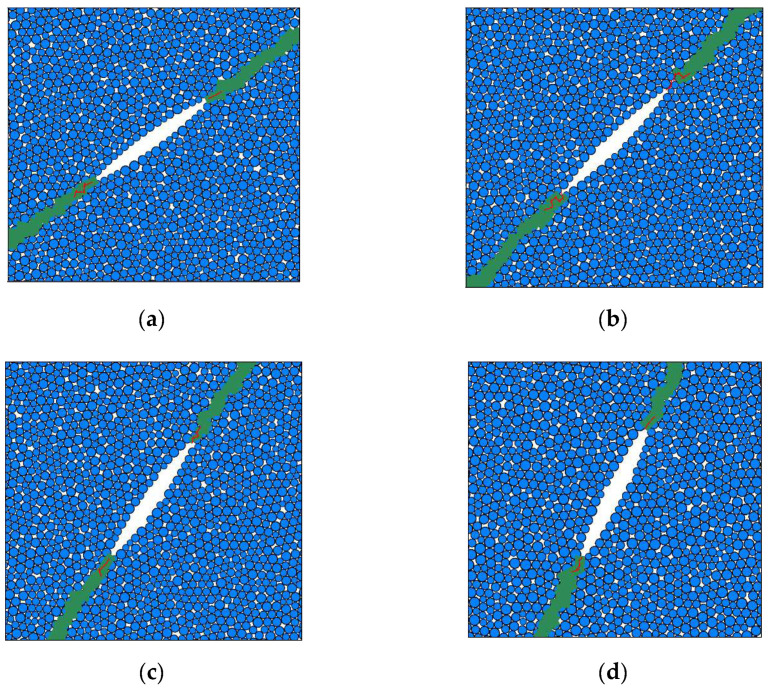
Initiation stress of interface crack with different joint inclination angles. (**a**) *β* = 35.8° (55.23 MPa), (**b**) *β* = 45° (28.28 MPa), (**c**) *β* = 54.2° (19.50 MPa), (**d**) *β* = 63.4° (20.01 MPa).

**Figure 9 materials-15-04881-f009:**
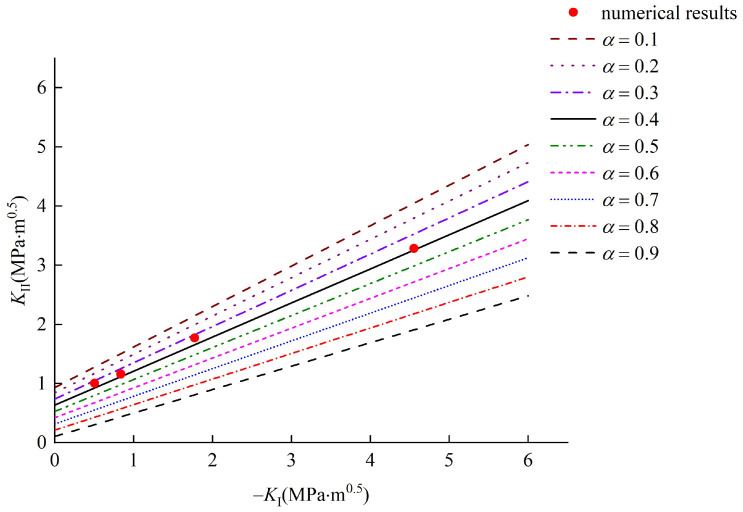
Comparison between the numerical results and the theoretical results.

**Figure 10 materials-15-04881-f010:**
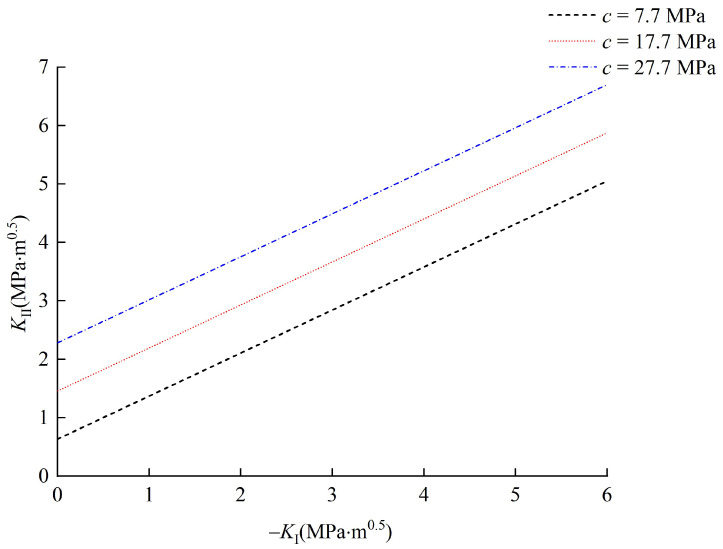
Effect of c on the initiation of interface crack.

**Figure 11 materials-15-04881-f011:**
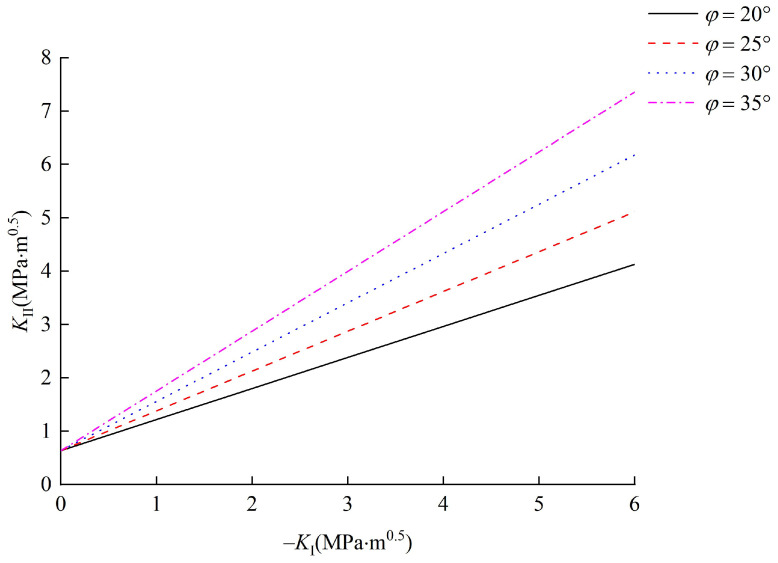
Effect of *φ* on the initiation of interface crack.

**Figure 12 materials-15-04881-f012:**
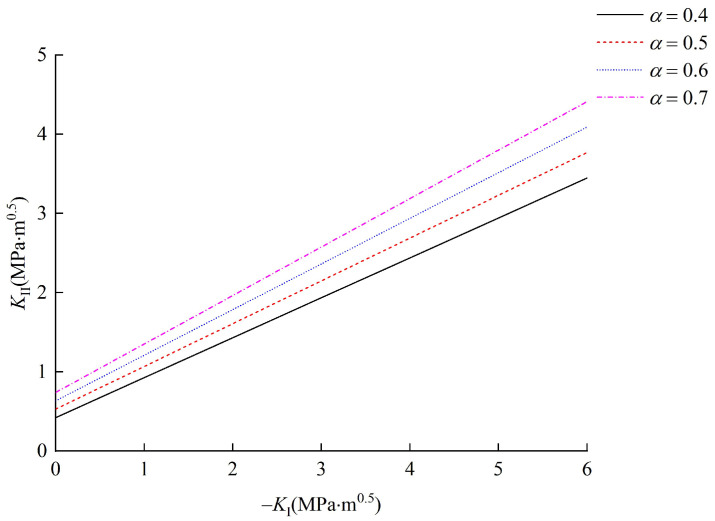
Effect of α on the initiation of interface crack.

**Table 1 materials-15-04881-t001:** Micro-parameters of the numerical model.

Micro-Parameters	Symbol	Units	Value
Minimum particle radius	*R* _min_	mm	0.17
Ratio of maximum to minimum ball radius	*λ*	/	1.76
Ratio of normal to shear stiffness of the particle	*k*_n_/*k*_s_	/	1.6
Ratio of normal to shear stiffness of the parallel bond	*k*_n_/*k*_s_	/	1.6
Young’s modulus of the particle	*E* _c_	GPa	7.3
Young’s modulus of the parallel bond	*E* _c_	GPa	7.3
Particle friction coefficient	*μ*	/	0.611
Particle density	*ρ*	kg/m^3^	2400
Parallel-bond tensile strength	*σ* _b_	MPa	30.0
Parallel-bond cohesion	*c* _b_	MPa	50.0
Smooth-joint particle friction coefficient	*μ* _sj_	/	0.36
Smooth-joint tensile strength	*σ* _sj_	MPa	6.0
Smooth-joint cohesion	*c* _sj_	MPa	7.7

**Table 2 materials-15-04881-t002:** Stress intensity factors of interface crack with different inclination angles.

Dip Angle	35.8°	45°	54.2°	63.4°
*K*_I_ (MPa‧m^0.5^)	−4.55	−1.77	−0.83	−0.50
*K*_II_ (MPa‧m^0.5^)	3.28	1.77	1.16	1.01

**Table 3 materials-15-04881-t003:** Comparison with other traditional theories.

Fracture (Initiation) Criterion	The MTS Criterion [21]	The MTS Criterion Considering T-Stress [45]	The Proposed Criterion
Application scope	Cracks in rock matix	Cracks in rock matrix	Cracks in rock joint
Fracture mode	Mode I fracture	Mode I fracture	Mode II fracture
Initiation condition	$K_{I\max}/K_{IC}=1$	$K_{I\max}/K_{IC}=1$	$K_{II\max}/K_{IIC}=1$
Key parameters	*K*_I_, *K*_II_	*K*_I_, *K*_II_, *α*, *r_c_*, *E* ^1^, *ν* ^2^, *β*, *f* ^3^, *k_n_* ^4^, *k_s_* ^5^	*K*_I_, *K*_II_, *φ*, *c*, *α*, *r_c_*
Initiation angle	70.5°	Variable	0°

^1^ *E* is the Young’s elastic modulus of the rock matix. ^2^ *ν* is the Poisson’s ratio of the rock matix. ^3^ *f* is the friction coefficient of the crack surfaces. ^4^ *k_n_* is the normal stiffness of the crack surfaces. ^5^ *k_s_* is the shear stiffness of the crack surfaces.

## Data Availability

The raw/processed data required to reproduce these findings cannot be shared at this time as the data also forms part of an ongoing study.

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
