# Peer review of "Study on the Initiation of Interface Crack in Rock Joints"

_materials, 2022, doi:10.3390/ma15144881_

Round 1
Reviewer 1 Report
Reviewer comments:
I see that the work was very well designed and executed. I can comment with only some minor points.
1. There are so many English grammatical errors in the manuscript. Also, any sentence cannot start with “And” as provided in the abstract, please revise. The authors need to revise the language throughout the manuscript.
2. The information provided in the Introduction section is very poor. The authors need to provided updated information and has to be at the highest scientific level. It should cover the background, current techniques, rationale etc.
3. The discussion section too needs improvement.
Reviewer 2 Report
Dear authors,
This paper presents a model to solve the interfacial fracture of rock joints. Some interesting results are obtained. It is fit for publication with the following major corrections:
-Check the current template and the guidelines of the Materials journal.
-The title refers to STUDY, but the work is about MODELING.
-In general, the manuscript needs a grammar revision since several mistakes were observed along it.
-Review the phrasing of the introduction. It is somewhat confusing.
-Add the potential use of the proposed model. For geotechnics, power generation, oil wells, etc.
-Avoid the use of can't, aren't, tests'
-The value of R^2 is questionable for Figure 4.
-The new fracture criterion should be corroborated with a larger number of experimental results or simulation conditions.
-If the initial stress state were different, does the model still work?
-According to the developed model, what is the research perspective?
Reviewer 3 Report
The manuscript is not in the required journal template.
Section 4.1 including photo and graph is an outcome of some other study. Why it has to be discussed here?
The entire discussions (up to section 5) are barely taken from previous articles. If so, what is the role of this paper?
The experimental study needs to be carried out to support the model developed rather than mentioning “it is difficult to conduct the test in the laboratory using the interface crack specimens, to verify the proposed fracture criterion of interface crack comprehensively and deeply”.
“To study the effect of cohesion c, the parameter c is taken as 7.7 MPa, 17.7 MPa and 27.7 MPa, respectively, and the other parameters, such as internal friction angle φ and relative critical size α, are fixed as 20° and 0.6, respectively” Why particularly?
“To study the effect of friction angle φ, the parameter φ is taken as 20°, 25°, 30° and 35°, respectively, and the other parameters, such as cohesion c and relative critical size α, are fixed as 7.7 MPa and 0.6, respectively.” Why particularly?
“It can be seen from Fig. 11 that the relationship curves between Mode Ι and ΙΙ stress intensity factors are monotonically increasing, and as the increase of internal friction angle φ, the slope of the curves enlarge.” Why?
“To study the effect of relative critical size α, the parameter α is taken as 0.4, 0.5, 0.6 and 0.7, respectively, and the other parameters, such as cohesion c and internal friction angle φ, are fixed as 7.7 MPa and 20°, respectively.” Why particularly?
“It can be seen from Fig. 12 that the relationship curves between Mode Ι and ΙΙ stress intensity factors are monotonically increasing too and with the increase of relative critical size α, both the slope and intercept of the curves enlarge.” Any justification?
“The theoretical results calculated by the proposed fracture criterion are agreement well with the experimental and numerical results. That is to say, the proposed fracture criterion can be verified by the tests' results well”. When mentioned already as “it is difficult to conduct the test in the laboratory using the interface crack specimens, to verify the proposed fracture criterion of interface crack comprehensively and deeply”, how this conclusion can be accepted?
Justification for the third conclusion?
Reviewer 4 Report
The paper Study on initiation of interface crack in rock joints
Authors Xin Chen, Wei Gao*, Shuangshuang Ge and Cong Zhou presents an interesting topic to Materials readers, but some corrections are necessary.
My principal questions or remarks:
The manuscript isn’t in journal format.
The citations aren’t in journal format…. The authors must use [number].
The title is clear.
The content is in accord with title.
The manuscript adheres to the journal's standards after revision.
The size of the article is appropriate to the contents.
The authors underlined the major findings of their work and explain novelty of this study.
The Abstract must be revised. The Abstract section refers to the objectives, novelty, study findings, methodologies, discussion as well as conclusion.
The key words permit found article in the current registers or indexes.
In the introduction it isn’t clearly described the state of the art of the investigated problem. Please cited articles from last years if this study is actual.
The methods are well described.
The figures have a good quality.
The tables contain necessary results.
Please provide comparison with other studies. It is necessary, in tabular form, to provide the comparison studies.
The Conclusion must be revised.
The paper was written in standard, grammatically correct English, small corrections are necessary.
The references aren’t in journal’s format. Please respect guide for authors.
If the paper is in Materials journal topics, please provide 2 references from this journal (last year).
The paper is relatively easy to understand by readers from other area.
Please provide:
Author Contributions: For research articles with several authors, a short paragraph specifying their individual contributions must be provided. The following statements should be used “Conceptu-alization, X.X. and Y.Y.; methodology, X.X.; software, X.X.; validation, X.X., Y.Y. and Z.Z.; formal analysis, X.X.; investigation, X.X.; resources, X.X.; data curation, X.X.; writing—original draft preparation, X.X.; writing—review and editing, X.X.; visualization, X.X.; supervision, X.X.; project administration, X.X.; funding acquisition, Y.Y. All authors have read and agreed to the published version of the manuscript.” Please turn to the CRediT taxonomy for the term explanation. Au-thorship must be limited to those who have contributed substantially to the work reported.
Funding: Please add: “This research received no external funding” or “This research was funded by NAME OF FUNDER, grant number XXX” and “The APC was funded by XXX”. Check carefully that the details given are accurate and use the standard spelling of funding agency names at https://search.crossref.org/funding. Any errors may affect your future funding.
Institutional Review Board Statement: In this section, you should add the Institutional Review Board Statement and approval number, if relevant to your study. You might choose to exclude this statement if the study did not require ethical approval. Please note that the Editorial Office might ask you for further information. Please add “The study was conducted in accordance with the Declaration of Helsinki, and approved by the Institutional Review Board (or Ethics Committee) of NAME OF INSTITUTE (protocol code XXX and date of approval).” for studies involving humans. OR “The animal study protocol was approved by the Institutional Review Board (or Ethics Com-mittee) of NAME OF INSTITUTE (protocol code XXX and date of approval).” for studies involving animals. OR “Ethical review and approval were waived for this study due to REASON (please provide a detailed justification).” OR “Not applicable” for studies not involving humans or animals.
Informed Consent Statement: Any research article describing a study involving humans should contain this statement. Please add “Informed consent was obtained from all subjects involved in the study.” OR “Patient consent was waived due to REASON (please provide a detailed justification).” OR “Not applicable.” for studies not involving humans. You might also choose to exclude this statement if the study did not involve humans.
Written informed consent for publication must be obtained from participating patients who can be identified (including by the patients themselves). Please state “Written informed consent has been obtained from the patient(s) to publish this paper” if applicable.
Data Availability Statement: In this section, please provide details regarding where data sup-porting reported results can be found, including links to publicly archived datasets analyzed or generated during the study. Please refer to suggested Data Availability Statements in section “MDPI Research Data Policies” at https://www.mdpi.com/ethics. If the study did not report any data, you might add “Not applicable” here.
Acknowledgments: In this section, you can acknowledge any support given which is not covered by the author contribution or funding sections. This may include administrative and technical support, or donations in kind (e.g., materials used for experiments).
Conflicts of Interest: Declare conflicts of interest or state “The authors declare no conflict of in-terest.” Authors must identify and declare any personal circumstances or interest that may be perceived as inappropriately influencing the representation or interpretation of reported research results. Any role of the funders in the design of the study; in the collection, analyses or interpretation of data; in the writing of the manuscript; or in the decision to publish the results must be declared in this section. If there is no role, please state “The funders had no role in the design of the study; in the collection, analyses, or interpretation of data; in the writing of the manuscript; or in the decision to publish the results”.
Round 2
Reviewer 2 Report
Manuscript is well revised by the authors and all the corrections and suggestions are incorporated in the revised manuscript, so it is accepted in the current format by my side.
Reviewer 3 Report
Authors have carried out the comments.
Reviewer 4 Report
The manuscript was improved in accord with recommendations.